# Training–Inference Consistent Segmented Execution for Long-Context LLMs

Xianpeng Shang [1 2 3]   Jiang Li [1 2 3]   Zehua Duo [1 2 3]   Qianyi Cai [4]   Xiangdong Su [1 2 3]

## Abstract

Transformer-based large language models face severe scalability challenges in long-context generation due to the computational and memory costs of full-context attention. Under practical computation and memory constraints, many inference-efficient long-context methods improve efficiency by adopting bounded-context or segment-level execution only during inference, while continuing to train models under full-context attention, resulting in a mismatch between training and inference execution and state-transition semantics. Based on this insight, we propose a training–inference consistent segment-level generation framework, in which training and inference follow the same segment-level forward execution semantics. During training, consistency with inference is enforced by restricting gradient propagation to KV states carried over from the immediately preceding segment, while permitting head-specific access to past KV states during the forward pass without involving them in gradient propagation. Across long-context benchmarks, our approach achieves performance comparable to full-context attention, while achieving competitive latency–memory trade-offs against strong inference-efficient baselines, and substantially improving scalability at very long context lengths (e.g., approximately $6\times$ lower peak prefill memory at 128K compared to full-context attention with FlashAttention).

[1]College of Computer Science, Inner Mongolia University, Hohhot 010021, China [2]National & Local Joint Engineering Research Center of Intelligent Information Processing Technology for Mongolian, Hohhot 010021, China [3]Inner Mongolia Key Laboratory of Multilingual Artificial Intelligence Technology, Hohhot 010021, China [4]Thrust of Artificial Intelligence, The Hong Kong University of Science and Technology (Guangzhou), China. Correspondence to: Xiangdong Su <cssxd@imu.edu.cn>.

*Proceedings of the $43^{rd}$ International Conference on Machine Learning*, Seoul, South Korea. PMLR 306, 2026. Copyright 2026 by the author(s).

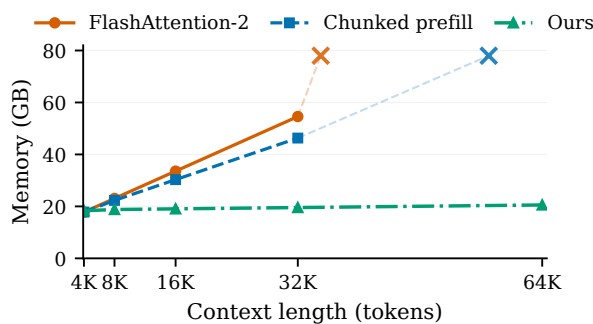

*Figure 1.* Peak GPU memory consumption during long-context prefill.

## 1. Introduction

Long-context modeling is increasingly vital for large language models (Achiam et al., 2023; Team et al., 2024; Anthropic, 2024), underpinning practical applications like document understanding, sustained dialogue, and complex reasoning (Bai et al., 2024). However, the quadratic computational complexity of full-context self-attention fundamentally limits the scalability of Transformer models in long-context scenarios (Vaswani et al., 2017; Keles et al., 2023). Accordingly, long-context inference is often performed using restricted execution regimes, such as bounded-context or chunked attention, to reduce computational costs (Xiao et al., 2024; Liu et al., 2025). Recent work has substantially improved the efficiency of long-context inference through execution-level optimizations that preserve attention semantics, reducing memory consumption and practical inference costs without altering model outputs (Dao, 2024; Agrawal et al., 2023). Yet, as context length continues to scale, the resource savings offered by such semantics-preserving execution-level optimizations alone are often insufficient in practice (Figure 1), and more restrictive execution strategies, such as windowed or sparse attention mechanisms, are commonly adopted.

Despite their effectiveness at inference time, most existing approaches impose these restricted execution regimes only during inference, while still relying on full-context attention during training. This leads to a mismatch between training and inference in both execution and cross-segment state evolution. As a result, the model can depend on information available during training that is absent under the constrained

inference regime, which can undermine stability and generalization in long-context settings.

To address this limitation, we propose a training–inference consistent segment-level generation framework, which treats segmented execution as a shared modeling assumption rather than an inference-time optimization. We partition the sequence into segments and carry only a fixed-size KV tail as the *sole* differentiable cross-segment interface state, which is used *verbatim* in both training and inference. Training restricts cross-segment credit assignment to the most recent $K$ state transitions via TBPTT; under this strictly bounded recursion, TBPTT computes the exact gradient of an inference-consistent objective, preventing reliance on information unavailable at inference time. To access evidence beyond the carried-KV horizon, the model additionally consumes a retrieved KV prefix in a forward-only manner (no gradient), which does not participate in state recursion. Architecturally, we realize this design with head- and layer-sparse long-range heads, while the majority of heads support local, state-carrying computation, yielding execution semantics that are strictly aligned between training and inference.

Our contributions are as follows:

- We propose a segment-level modeling framework for long-context modeling that enforces training–inference consistency by design, decomposing cross-segment information flow into a local continuity channel and a separate forward-only mechanism that provides long-range conditioning.

- We show that training–inference alignment can be theoretically guaranteed without introducing persistent memory variables, by restricting cross-segment learning to a controlled interface state. Under this constrained formulation, truncated backpropagation computes the exact gradient of the inference-consistent objective, rather than an approximation.

- We empirically validate the proposed training–inference consistent framework across multiple long-context benchmarks and context lengths, demonstrating strong performance under constrained execution, ablations indicating that TBPTT with $K{=}1$ is sufficient and optimal, and substantially improved scalability (e.g., $\sim 6\times$ lower memory at 128K-context prefill compared to full attention).

## 2. Related Work

**Execution-Level Optimizations.** Recent work has improved the efficiency of exact attention-based long-context inference in Transformer-based models through execution-

and system-level optimizations, without modifying model parameters, training objectives, or attention semantics. Representative examples include kernel-level optimizations such as FlashAttention and FlashAttention-2 (Dao et al., 2022; Dao, 2024), runtime systems for efficient KV cache management and scheduling such as vLLM and SARATHI (Kwon et al., 2023; Agrawal et al., 2023), as well as system-level approaches that explore heterogeneous memory offloading for large-scale inference, e.g., Flex-Gen (Sheng et al., 2023). While these methods reduce constant factors and improve throughput under moderate context lengths, they preserve full-context attention semantics and therefore do not address the computational and memory challenges encountered at much longer context lengths, where restricted execution is required.

**Restricted Execution at Inference.** Another line of work enables long-context inference by restricting attention connectivity or retained states only at inference time, while leaving training-time execution unchanged. Streaming-based methods, such as StreamingLLM (Xiao et al., 2024) and LM-Infinite (Han et al., 2024) achieve zero-shot length generalization by adapting the inference-time attention pattern of models trained on short contexts, while leaving training-time execution unchanged. In a different direction, MInference (Jiang et al., 2024) accelerates long-context prefill through inference-time sparse attention, while preserving dense attention during training. Restricted execution can also arise from state-level constraints rather than attention sparsification, as in ChunkKV (Liu et al., 2025), which selectively compresses and retains KV states during inference without retraining. Although restrictions are imposed in different ways, spanning attention scope, sparsity, and state retention, these approaches rely on training-time execution assumptions that differ from inference-time behavior, leading to mismatches in attention connectivity or retained-state usage at inference time.

**Training–Inference Alignment.** Training–inference alignment has been explored for settings where inference adopts restricted execution by explicitly incorporating comparable constraints during training. Longformer (Beltagy et al., 2020) replaces full self-attention with a fixed sparse pattern so that training and inference operate under identical attention connectivity. Core Context Aware (CCA) (Chen et al., 2025) enforces alignment under a reduced-context computation graph by applying the mechanism consistently during adaptation. Alignment has also been studied for streaming or segmented execution: Shiftable Context (Raffel et al., 2023) enforces consistent segment structures between training and inference, while Sliding Window Attention Training (Fu et al., 2025) trains models directly with windowed attention to avoid degradation when such constraints are introduced only at inference time. Taken together, these approaches mitigate training–inference mismatch by enforcing

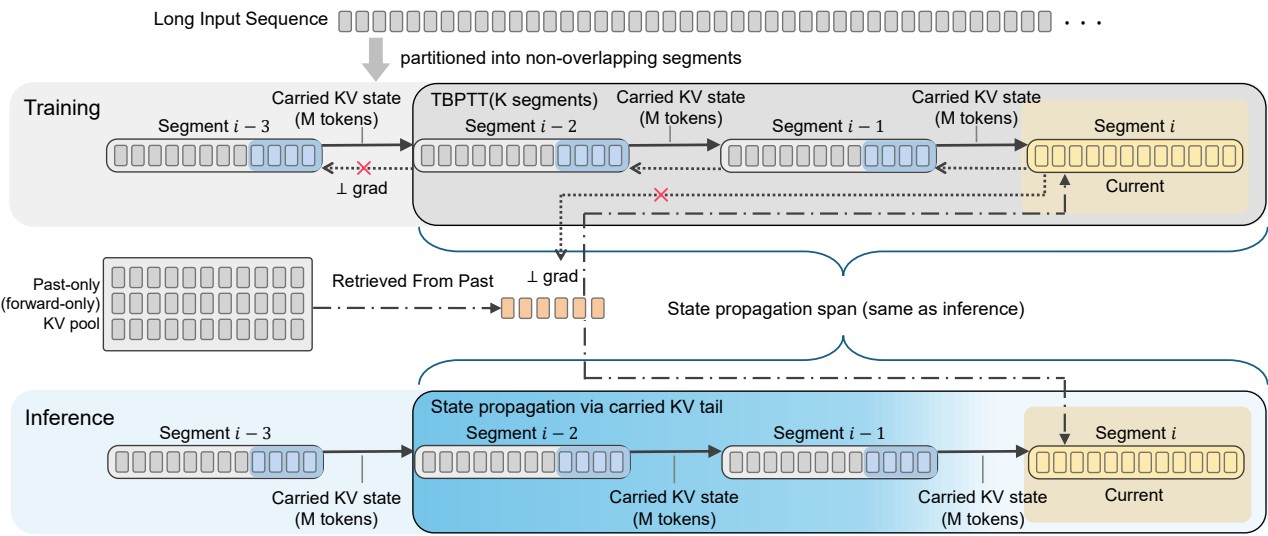

*Figure 2.* Training–inference consistent segmented execution. A sequence is processed segment by segment with two cross-segment inputs: a *carried KV tail* $C_{i-1}$ (the only *differentiable* state that propagates across segments) and an optional retrieved prefix $R_{i-1}$ read from a *past-only* KV pool. During training, TBPTT with depth $K$ truncates credit assignment along the state chain (red cross), so gradients flow through $C$ for at most $K$ segment transitions (blue brace), while the retrieval path and earlier history are forward-only (no gradient).

consistent attention connectivity or context structure across training and inference stages.

**Memory-Based and Recurrent Models.** An alternative modeling paradigm extends effective context length by introducing persistent memory mechanisms or recurrent state propagation across segments. Transformer-XL (Dai et al., 2019) is the earliest work to introduce segment-level recurrence with gradient truncation in Transformers, whereas our approach formalizes training–inference consistent execution semantics and separates short-term differentiable state propagation from forward-only long-range retrieval. Compressive Transformer (Rae et al., 2020) builds on Transformer-XL by retaining older states via learned compression. Recurrent Memory Transformer (Bulatov et al., 2022) further introduces explicit memory tokens that are recurrently updated and trained to store global information across segments. Memorizing Transformers (Wu et al., 2022) instead rely on an external key–value memory accessed via retrieval to support long-range recall. These methods rely on explicit, persistent cross-segment states to carry long-range information, whose update dynamics during training are not necessarily aligned with the execution semantics encountered at inference time. By contrast, our approach avoids persistent memory states and enforces training–inference consistency under segment-level execution.

## 3. Method

In long-context language modeling, the computational and memory costs of full-context attention make unrestricted execution impractical at scale, leading to the use of segmented

or otherwise constrained attention mechanisms. Many existing methods impose such constraints only at inference time, while retaining full-context attention during training, resulting in a mismatch between training and inference execution semantics. In this section, we describe a training–inference consistent segmented execution framework, in which both training and inference follow an identical segment-level execution scheme, explicitly constraining cross-segment state propagation and long-range access under the same execution semantics.

### 3.1. Training–Inference Consistent Segmented Execution

**Setup** As illustrated in Figure 2, we partition a token sequence into $N$ non-overlapping segments $\{x^{(i)}\}_{i=1}^{N}$, where $m_i = |x^{(i)}|$ denotes the length of segment $i$. To enable segmented inference under bounded attention, we expose a restricted cross-segment interface state $C_i \in \mathcal{C}$ (a fixed-size carried KV interface in our implementation), together with a forward-only retrieval prefix $R_i \in \mathcal{R}$ provided by a long-range module.

**Definition 3.1** (Segment-level execution semantics). For each segment $i$, the model runs the same forward operator at both training and inference:

$$(C_i, o^{(i)}) = F_\theta\big(x^{(i)}, C_{i-1}, R_{i-1}\big), \qquad (1)$$

where $o^{(i)}$ denotes outputs used to compute the LM loss, $C_{i-1}$ is the *only* differentiable cross-segment interface state available to the model, and $R_{i-1}$ is a forward-only retrieval

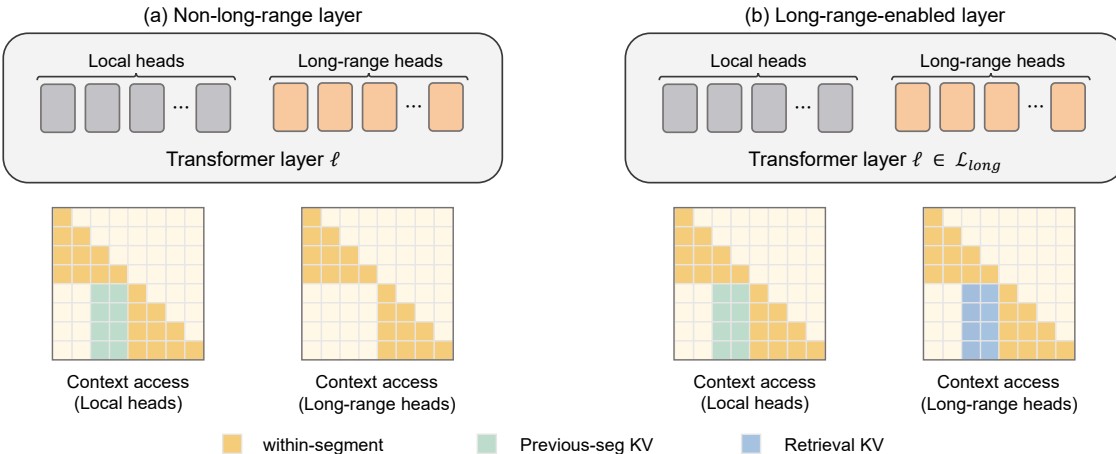

*Figure 3.* Head- and layer-sparse long-range retrieval. (a) In a non-long-range layer $\ell \notin \mathcal{L}_{\text{long}}$, local heads attend to within-segment tokens and the carried KV state from the previous segment (green), while long-range heads use within-segment causal attention only (orange). (b) In a long-range-enabled layer $\ell \in \mathcal{L}_{\text{long}}$, local heads remain unchanged, while long-range heads additionally attend to a retrieved prefix from a past-only KV pool (blue). In all cases, attention within the current segment remains causal.

prefix. The segment loss is

$$\ell_i(\theta; C_{i-1}, R_{i-1}) = -\sum_{t=1}^{m_i} \log p_\theta\left(x_t^{(i)} \mid x_{<t}^{(i)}, C_{i-1}, R_{i-1}\right). \tag{2}$$

**Operational interpretation.** For a sequence split into three segments $x^{(1)}, x^{(2)}, x^{(3)}$, with $C_0 = R_0 = \emptyset$, the segment-level recurrence in Eq. (1) unrolls as

$$(C_1, o^{(1)}) = F_\theta(x^{(1)}, \emptyset, \emptyset),$$
$$(C_2, o^{(2)}) = F_\theta(x^{(2)}, C_1, R_1),$$
$$(C_3, o^{(3)}) = F_\theta(x^{(3)}, C_2, R_2).$$

Here $o^{(i)}$ is used to predict tokens in the current segment $x^{(i)}$, while $C_i$ is the carried state produced for the next segment. Thus, $C_{i-1}$ and $R_{i-1}$ are conditioning inputs in Eq. (2), not prediction targets. Past tokens are not replaced by $C$ or $R$; rather, the model accesses them through two prefix inputs: $C$ carries recent local KV states from the previous segment, while $R$ provides forward-only retrieved KV states from earlier segments. Section 3.4 describes how these two inputs are implemented.

**Training–Inference Forward Identity** Eq. (1) is used *verbatim* at training and inference. The only difference lies in how gradients are allowed to propagate across the state chain, formalized next.

### 3.2. Inference-Consistent Truncated Training Objective

Under the segment-level execution semantics defined in Section 3.1, we next formalize the training objective induced by truncated state propagation.

**Definition 3.2** ($K$-truncated consistent objective). Let $\text{sg}(\cdot)$ denote the stop-gradient operator (identity in forward, zero gradient in backward). Let $b_i = \max(0, i - K - 1)$ denote the truncation boundary, with $C_0 = \emptyset$. For each $i$, define a truncated state chain $\tilde{C}_{i-1}^{(K)}$ by unrolling the state update for at most $K$ segments, with the boundary state $C_{b_i}$ treated as a constant:

$$\tilde{C}_{b_i}^{(K)} = \text{sg}(C_{b_i}), \tag{3}$$
$$\tilde{C}_j^{(K)} = \Phi_\theta\left(x^{(j)}, \tilde{C}_{j-1}^{(K)}, R_{j-1}\right), \quad j = b_i + 1, \ldots, i - 1. \tag{4}$$

where $\Phi_\theta$ is the state-update component induced by $F_\theta$. We define the consistent objective

$$L_K(\theta) = \sum_{i=1}^{N} \ell_i\left(\theta; \tilde{C}_{i-1}^{(K)}, R_{i-1}\right). \tag{5}$$

**Operational meaning of the truncation depth.** Continuing the three-segment unrolling above, consider the representative loss on the third segment, $\ell_3(\theta; \tilde{C}_2^{(K)}, R_2)$. With $K = 1$, gradients propagate through the state update that produces $C_2$ from segment 2 and stop at $C_1$. With $K = 2$, the state chain is further unrolled through the update that produces $C_1$ from segment 1, and gradients stop at $C_0$. In both cases, the forward computation remains unchanged and follows Eq. (1); Eqs. (3)–(4) only determine how far gradients propagate along the carried-state chain.

### 3.3. Exactness of TBPTT for the Stated Objective

**Theorem 3.3** (Exactness of TBPTT for $L_K$). *Truncated backpropagation through time (TBPTT) with truncation depth $K$ computes the exact gradient $\nabla_\theta L_K(\theta)$.*

*Proof sketch.* By Definition 3.2, the stop-gradient boundary $\mathrm{sg}(C_{b_i})$ removes all gradient paths across more than $K$ state transitions in the computational graph of each loss term. TBPTT with truncation depth $K$ performs backpropagation on exactly this truncated computational graph, therefore it yields $\nabla_\theta L_K(\theta)$. A detailed chain-rule derivation (Jacobian product / adjoint recursion) is provided in Appendix A (see Appendix A.2). □

**Corollary 3.4** (Training–inference alignment)**.** *Since training and inference share identical forward execution in Eq. (1), and TBPTT computes the exact gradient of the inference-consistent objective in Eq. (5), the resulting training procedure is fully aligned with inference under the same execution semantics.*

**Forward-Only Long-Range Retrieval**   In our system, the long-range module constructs $R_{i-1}$ from detached past KV states and uses it only in the forward pass, so it does not introduce additional cross-segment credit assignment paths. We formalize this as Lemma B.1 in Appendix B.

## 3.4. Model Architecture: Head- and Layer-Sparse Long-Range Retrieval

In Sections 3.1–3.3, we defined a training–inference consistent segmented execution semantics and proved that TBPTT computes the exact gradient of the corresponding consistent objective. We now instantiate the abstract operator $F_\theta(\cdot)$ in Eq. (1) with a Transformer decoder architecture that (i) implements the differentiable cross-segment interface state $C_i$ via a fixed-size carried KV tail, and (ii) optionally augments context using a forward-only retrieval prefix $R_i$ in a sparse manner across layers and heads. Figure 3 illustrates the resulting context-access patterns.

**Head and Layer Partition**   Consider an $L$-layer Transformer decoder with $H$ attention heads per layer. For each layer $\ell$, we partition heads into two disjoint groups: local heads $\mathcal{H}_{\mathrm{local}}$ and long-range heads $\mathcal{H}_{\mathrm{long}}$, and denote $\alpha := |\mathcal{H}_{\mathrm{local}}|/H$. This head-level sparsity follows observations from recent mechanistic and systems analyses showing that only a small subset of heads exhibits retrieval-like long-context behavior, whereas the majority behaves more like streaming/local heads (Wu et al., 2025; Xiao et al., 2025). We further select a small subset of layers $\mathcal{L}_{\mathrm{long}} \subseteq \{1, \ldots, L\}$ as long-range-enabled layers, where long-range heads additionally consume a retrieved prefix. At the layer level, prior work reports structured redundancy in Transformer attention layers, indicating that many attention layers can be removed with only limited performance degradation (He et al., 2024). In our default configuration, we enable long-range retrieval on layers $\mathcal{L}_{\mathrm{long}} = \{6, 8, 11, 18\}$ and use a prior-based long-range head group $\mathcal{H}_{\mathrm{long}} = \{0, 1, 2, 4, 9, 12, 14, 15, 16, 18, 19, 22, 23, 26, 29, 30\}$, with

the remaining heads assigned to $\mathcal{H}_{\mathrm{local}}$. We ablate alternative head groupings in Appendix G.6. All layer and head indices follow the zero-indexed implementation convention.

**Local Continuity Channel: Carried KV Interface** $\{C_i\}$
For each segment $x^{(i)}$, after computing per-layer key/value states, we form the cross-segment interface state $C_i$ as a fixed-size carried KV tail of length $M$ produced by local heads. Concretely, for each layer $\ell$, we cache the last $M$ key/value vectors from $\mathcal{H}_{\mathrm{local}}$ and expose them to the next segment as the only differentiable cross-segment state. When processing segment $i$, local heads attend to the concatenation of this carried KV tail (from segment $i-1$) and the within-segment KV of segment $i$ (Figure 3, green+orange). This realizes the interface state $C_{i-1}$ used in Eq. (1), and is exactly the state channel whose gradient propagation is aligned with inference (Corollary 3.4).

**Long-Range Channel: Forward-Only Retrieval Prefix** $\{R_i\}$   To capture dependencies beyond the carried KV horizon, we introduce an external past-only KV pool used by long-range-enabled layers. For each $\ell \in \mathcal{L}_{\mathrm{long}}$, we maintain a past-only pool that stores historical key/value vectors computed from long-range heads and supports retrieving a fixed-size prefix of length $R$ via top-$k$ similarity search. In our implementation, we do not apply eviction; the pool retains historical KV states for the selected long-range heads and long-range-enabled layers. Before processing segment $i$, we construct a compact query summary from the tail of segment $i-1$ (using long-range heads) and retrieve $R$ key/value vectors from the pool as the retrieval prefix $R_{i-1}$. In layers $\ell \in \mathcal{L}_{\mathrm{long}}$, long-range heads attend to the concatenation of this retrieved prefix and the within-segment KV of segment $i$ (Figure 3, blue+orange). In layers $\ell \notin \mathcal{L}_{\mathrm{long}}$, long-range heads simply reduce to standard within-segment causal attention only (Figure 3, orange). KV states inserted into the pool are detached for future retrieval, and retrieval itself is forward-only, ensuring consistency with Sections 3.1–3.3. Formal analysis and implementation details are provided in Lemma B.1 and Appendix D.

**RoPE Re-indexing for Prefix Concatenation**   Both the carried KV tail and the retrieval prefix are treated as a prefix preceding the current segment. To preserve correct positional encoding under this concatenation, we re-index RoPE positions by assigning prefix positions $\{0, \ldots, P-1\}$ (with $P = M$ for the carried KV tail and $P = R$ for the retrieval prefix) and shifting the current segment positions by an offset of $P$. This allows reuse of standard RoPE attention while integrating prefix KV as preceding context. We provide exact position assignments and the corresponding RoPE application details in Appendix E.

Finally, by enabling retrieval only for a subset of layers and

heads, the architecture preserves the local execution semantics used in Sections 3.1–3.3, while limiting the attention-visible compute and activation-memory overhead of long-range context access; we quantify the time/memory benefits in Section 3.5.

## 3.5. Computational Cost under Segment-Level Restricted Execution

We analyze the computational and memory cost of the proposed architecture (Section 3.4) under the segment-level execution semantics in Definition 3.1.

We consider a sequence of total length $T = \sum_{i=1}^{N} |x^{(i)}|$, where $\{x^{(i)}\}_{i=1}^{N}$ are the non-overlapping segments used in Eq. (1). For clarity, we assume a constant segment length $|x^{(i)}| = S$, hence $N = T/S$. The differentiable cross-segment interface state $C_i$ is a carried KV tail of fixed length $M$, and the forward-only retrieval prefix $R_i$ has a fixed length $R$. Each layer has $H$ attention heads, partitioned into local heads $\mathcal{H}_{\text{local}}$ and long-range heads $\mathcal{H}_{\text{long}}$ (Section 3.4), with $\alpha := |\mathcal{H}_{\text{local}}|/H$. Only layers in $\mathcal{L}_{\text{long}}$ consume the retrieval prefix; denote $\beta := |\mathcal{L}_{\text{long}}|/L$.

**Time Complexity.** For a segment of length $S$, local heads attend to a context of size $S + M$ (within-segment plus carried tail). Long-range heads attend to $S$ in layers $\ell \notin \mathcal{L}_{\text{long}}$, and to $S + R$ in layers $\ell \in \mathcal{L}_{\text{long}}$. Abstracting the dominant attention cost in each layer as $\mathcal{O}(S \cdot L_{\text{ctx}})$ per head, the effective context length per token can be summarized as

$$S + \alpha M + \beta(1 - \alpha)R.$$

Therefore, the total attention time over the full sequence is

$$\mathcal{O}\big(T \cdot \big(S + \alpha M + \beta(1 - \alpha)R\big)\big),$$

(up to a constant factor depending on the number of layers $L$ and heads $H$).

**Retrieval Overhead.** In addition to attention, retrieval incurs an extra search cost over the past-only KV pool. In our implementation, the pool is not capacity-bounded: it retains historical KV states only for the selected long-range heads and long-range-enabled layers. Consequently, the searchable pool grows with the processed history, but this growth is reduced by the sparsity factor $\beta(1 - \alpha)$ over layers and heads. This retrieval cost is separate from the attention cost above: after retrieval, each long-range head attends only to a fixed-size prefix of length $R$, so the attention-visible context remains bounded by $S + \alpha M + \beta(1 - \alpha)R$.

**Memory Complexity.** Under segment-level execution, the peak KV cache for attention is dominated by the current segment KV and the prefix KV. Across all layers, local heads attend to an attention-visible KV length of $S + M$.

For long-range heads, the attention-visible KV length is $S + R$ in layers $\ell \in \mathcal{L}_{\text{long}}$ and $S$ in layers $\ell \notin \mathcal{L}_{\text{long}}$. Thus, the peak attention-visible KV length can be summarized as

$$\bar{L}_{\text{KV}} = S + \alpha M + \beta(1 - \alpha)R,$$

implying a bounded per-segment attention-memory footprint that does not scale with $T$. The persistent retrieval pool is separate from this attention-visible KV footprint. Since we retain historical KV states for the selected long-range heads and long-range-enabled layers without eviction, the pool memory grows linearly with the processed sequence length, with an effective sparsity factor $\beta(1 - \alpha)$. Importantly, these stored states are forward-only and do not create additional gradient-carrying activation memory.

# 4. Experiments

## 4.1. Experimental Setup

**Benchmarks and Tasks.** We evaluate long-context modeling through a set of complementary evaluation views: (1) language modeling perplexity (PPL) across varying context lengths, which characterizes stability and degradation patterns as the effective context length increases; (2) downstream performance on the LongBench benchmark (Bai et al., 2024), which comprises a diverse set of long-context tasks requiring cross-segment information integration and long-range dependency modeling, providing a task-level evaluation of long-context generalization; and (3) controlled length-scaling analyses using the CWE and FWE tasks from the RULER benchmark (Hsieh et al., 2024). RULER offers a setting in which context length can be systematically extended, enabling focused stress testing of robustness and dependency modeling behavior.

**Models and Backbones.** Our primary experiments are conducted on LLaMA 2 models (Touvron et al., 2023) with extended context lengths of 32K and 80K (Fu et al., 2024), which are widely used settings in long-context evaluation. We also report results on LLaMA 3.1 for LongBench v2 (Bai et al., 2025) to verify whether similar trends hold on a more recent backbone, with detailed results provided in appendix G for completeness.

**Baselines.** We compare our method against representative long-context modeling approaches with different execution constraints. These include full self-attention as an unconstrained reference, Core Context Aware (CCA) (Chen et al., 2025) (training–inference aligned via context compression), MInference (Jiang et al., 2024) (inference-only sparse attention), StreamingLLM (Xiao et al., 2024) (sliding-window attention with sinks), and DuoAttention (Xiao et al., 2025) (head-level separation for selective long-range access).

**Implementation Details.** For methods whose execution requires training–inference alignment (our method and CCA),

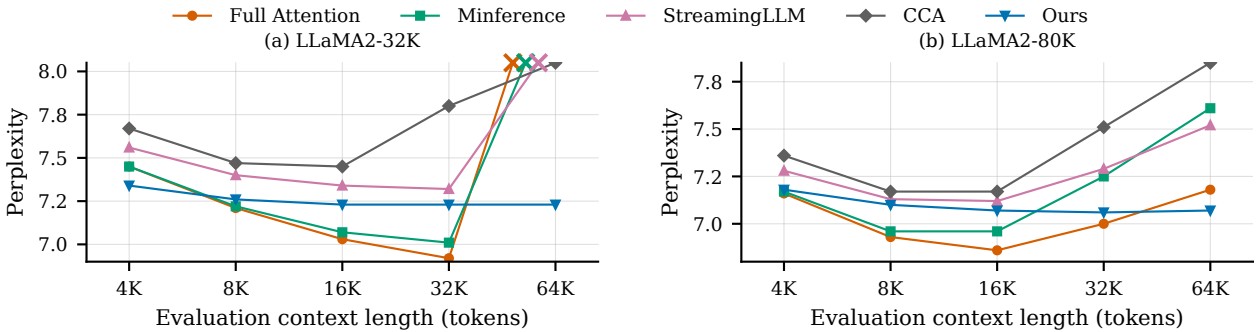

*Figure 4.* Perplexity on PG19 test under varying evaluation context lengths. Results are reported for LLaMA2-32K and LLaMA2-80K.

*Table 1.* Comparison of different methods on LongBench-E (Bai et al., 2024). TTFT (time to first token) measures the latency of prompt processing before token generation, while Mem. reports the peak GPU memory allocated during the prefill stage; lower values indicate better performance for both TTFT and Mem. TTFT and Mem. are measured with a 16K-token prefill for LLaMA2-7B-32K and a 32K-token prefill for LLaMA2-7B-80K. All experiments are conducted on A100 GPUs.

| Methods | S. QA | M. QA | Sum. | FS. Learning | Synthetic | Code | Avg. | TTFT (s) | Mem. (GB) |
|---|---|---|---|---|---|---|---|---|---|
| *LLaMA2-7B-32K (Vanilla Self-Attention)* | 2.82 | 6.09 | 3.09 | 65.36 | 0.50 | **60.91** | 23.13 | 1.62 | 23.61 |
| CCA-attention (Chen et al., 2025) | **3.36** | 5.64 | 4.60 | 55.41 | **2.12** | 55.56 | 21.12 | 1.79 | 28.08 |
| StreamingLLM (Xiao et al., 2024) | 1.93 | 5.22 | 3.95 | 61.93 | 0.25 | 58.12 | 21.90 | 1.59 | 22.19 |
| DuoAttention (Xiao et al., 2025) | 2.85 | 6.30 | 3.45 | 64.82 | 0.67 | 59.91 | 23.00 | **1.53** | **18.15** |
| MInference (Jiang et al., 2024) | 2.87 | **6.39** | 3.56 | **65.44** | 0.00 | 60.22 | 23.08 | 2.84 | 22.19 |
| (Ours) | 1.90 | 4.07 | **12.10** | 62.30 | 0.28 | 58.81 | **23.24** | 1.70 | 18.56 |
| *LLaMA2-7B-80K (Vanilla Self-Attention)* | 6.89 | 5.93 | 11.35 | **66.51** | 0.77 | 48.83 | 23.38 | 4.13 | 34.67 |
| StreamingLLM (Xiao et al., 2024) | 3.48 | 3.76 | 6.60 | 62.35 | 0.17 | 53.02 | 21.56 | **3.07** | 31.77 |
| CCA-attention (Chen et al., 2025) | 6.22 | 7.86 | 6.11 | 58.2 | **1.73** | 51.75 | 21.98 | 3.88 | 43.64 |
| DuoAttention (Xiao et al., 2025) | 6.57 | 6.12 | **11.86** | 63.68 | 0.89 | 48.50 | 22.94 | 3.79 | 23.66 |
| MInference (Jiang et al., 2024) | 6.93 | 5.96 | 11.19 | 66.42 | 1.01 | 48.60 | 23.35 | 4.13 | 31.77 |
| (Ours) | **7.58** | **8.56** | 8.70 | 63.76 | 0.42 | **56.02** | **24.17** | 3.49 | **19.06** |

we fine-tune the pretrained models using a standard language modeling objective to match their inference-time execution semantics. All alignment-based methods are fine-tuned under identical settings. All other baseline methods are evaluated under their standard pretrained configurations without additional training. For StreamingLLM, we adopt the implementation provided in MInference (Jiang et al., 2024). Additional implementation details are provided in the Appendix F. Our code is available at: link.

### 4.2. Main Results

**Perplexity under Increasing Context Lengths.** We evaluate perplexity on the PG19 test set under increasing evaluation context lengths. Figure 4 reports results for both LLaMA2-32K and LLaMA2-80K. Evaluation at 64K exceeds the training context length for LLaMA2-32K, and represents a challenging long-context regime in terms of modeling stability across methods.

Across commonly used medium-length contexts (4K–32K), different methods exhibit distinct perplexity trends as context length increases. Methods based on sparse, windowed, or compressed-context mechanisms (e.g., MInference, StreamingLLM, and CCA) often show non-monotonic

behavior with noticeable fluctuations. Full attention remains relatively smooth at moderate lengths but degrades as context length grows.

In contrast, our method shows a smoother perplexity trend across the evaluated context lengths, with perplexity increasing gradually as context length grows, and does not exhibit abrupt spikes when the evaluation length exceeds the training context.

**Downstream Performance on LongBench-E.** Table 1 reports results on LongBench-E across two long-context backbones. Overall, our method attains the highest average score under both settings, with improvements observed at both 32K and 80K across multiple task categories.

The largest improvement is observed on summarization, where our method outperforms all baselines at 32K (e.g., 12.1 vs. 3.09–4.60), while maintaining comparable performance on question answering tasks. At longer contexts, these gains extend to multi-document QA, where our method attains an average score of 8.56 at 80K, contributing to the highest overall average.

These results reflect the effect of enforcing training–inference consistency at the segment level, and correspond

*Table 2.* Performance as context length increases from 4K to 64K on RULER tasks (Hsieh et al., 2024). CWE and FWE denote the Common Words Extraction and Frequent Words Extraction tasks, respectively. Scores report recall-based accuracy at each context length. "–" indicates zero performance, i.e., failure to generalize to the corresponding length. Avg* reports the average over 4K–32K, excluding the 64K setting.

| Methods | 4K | | 8K | | 16K | | 32K | | 64K | | Avg* | |
|---|---|---|---|---|---|---|---|---|---|---|---|---|
| | CWE | FWE | CWE | FWE | CWE | FWE | CWE | FWE | CWE | FWE | CWE | FWE |
| *LLaMA2-7B-32K (Vanilla Self-Attention)* | 60.65 | 48.00 | 39.25 | 35.83 | 20.05 | 47.50 | 11.80 | 34.00 | – | – | 32.94 | 41.33 |
| StreamingLLM (Xiao et al., 2024) | 61.80 | 47.83 | 20.45 | 38.83 | 19.35 | 43.00 | 9.50 | 35.83 | – | – | 27.78 | 41.37 |
| MInference (Jiang et al., 2024) | 60.35 | 48.67 | 39.40 | 36.17 | 20.15 | 46.17 | 11.85 | 36.00 | – | – | 32.94 | 41.75 |
| CCA-attention (Chen et al., 2025) | 41.10 | 45.67 | 36.25 | 39.50 | 15.90 | 29.83 | 6.35 | 12.83 | – | – | 24.90 | 31.96 |
| DuoAttention (Xiao et al., 2025) | 60.15 | 48.83 | 38.80 | **42.67** | 18.15 | **48.33** | 12.20 | 33.83 | – | – | 32.33 | 43.42 |
| Ours | **70.60** | **53.83** | **54.15** | 38.67 | **41.45** | 44.83 | **19.35** | **38.17** | 2.00 | 34.17 | **46.39** | **43.88** |

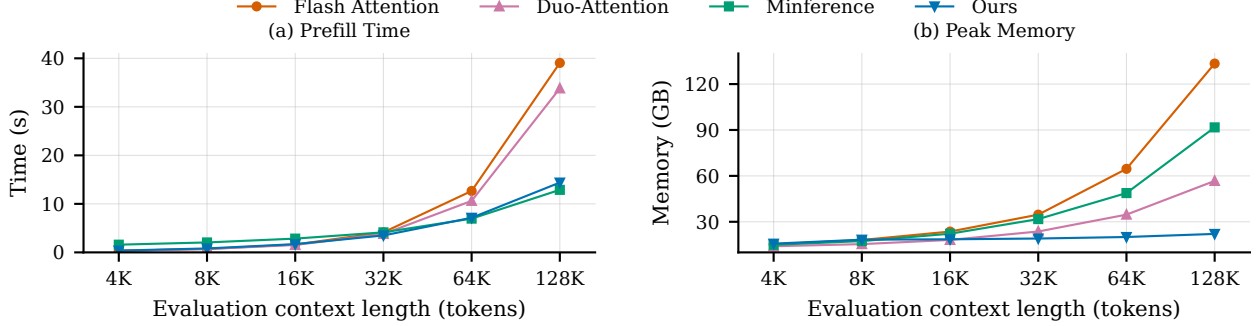

*Figure 5.* Prefill latency and peak GPU memory under increasing evaluation context lengths for LLaMA2-7B-32K. (a) Prefill time measures the forward-pass latency (seconds) to process the entire input prompt. (b) Peak memory (GB) reports the maximum GPU allocated memory during the prefill stage.

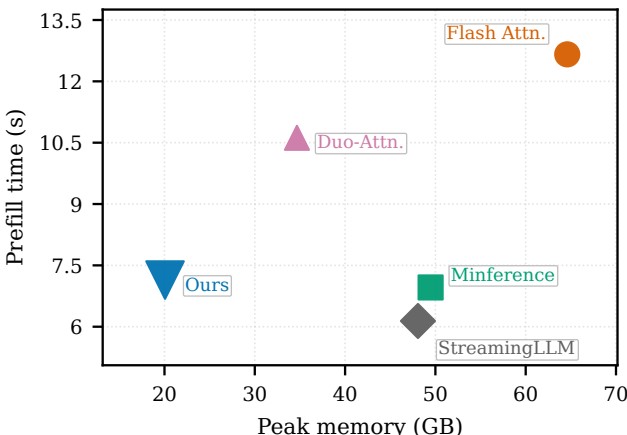

*Figure 6.* Prefill latency versus peak GPU memory at 64K context length for LLaMA2-7B-32K.

**Length Generalization on RULER.** We evaluate length generalization on the RULER benchmark by increasing the context length from 4K to 64K in Table 2. Within the effective context range of LLaMA2-7B-32K (4K–32K), performance degrades for all methods as context length grows; however, our method consistently achieves higher and more stable accuracy on both CWE and FWE, resulting in the best average performance (Avg*: 46.39 / 43.88) among all baselines. When the context length is further extended beyond the training range to 64K, most existing methods collapse to zero performance, whereas our method retains non-zero accuracy, exhibiting a smoother degradation behavior rather than an abrupt failure. These results indicate that under strictly constrained execution, our method demonstrates stronger robustness to context length scaling, including beyond the nominal training context range.

### 4.3. Prefill Latency and Memory Efficiency

We analyze efficiency in long-context settings by examining prefill latency and peak GPU memory consumption as the evaluation context length increases. As shown in Figure 5, while both latency and memory grow with context length for all methods, our approach exhibits substantially more gradual scaling behavior than existing baselines, particularly at longer contexts.

to improved robustness in long-context modeling across both local generation and cross-segment reasoning tasks. All improvements are obtained while preserving the efficiency advantages of segment-level execution, with lower memory usage and a favorable latency–memory trade-off compared to full attention. A detailed efficiency analysis is provided in Section 4.3. Results on a more recent backbone (LLaMA 3.1) and LongBench v2 are reported in Appendix G for completeness.

*Table 3.* Effect of training–inference alignment and TBPTT depth on LongBench-E category scores. The ablation evaluates the necessity of alignment and the impact of truncation depth.

| Method | S.QA | M.QA | Sum | FS. | Syn. | Code | Avg |
|---|---|---|---|---|---|---|---|
| Aligned (TBPTT=1) | 7.58 | 8.56 | 8.70 | 63.76 | 0.42 | 56.02 | **24.17** |
| Misaligned | 3.52 | 2.79 | 6.80 | 36.97 | 0.00 | 21.37 | 11.91 |
| Aligned (TBPTT=2) | 8.88 | 8.48 | 9.36 | 64.42 | 0.58 | 52.71 | 24.07 |

Figure 6 further illustrates this effect at a representative context length of 64K. Our method maintains a low peak memory footprint and moderate prefill latency, whereas other methods either incur significantly higher memory usage or suffer from increased latency. Overall, these results indicate that our approach achieves a more balanced latency–memory trade-off during long-context prefill, rather than optimizing a single efficiency dimension in isolation.

### 4.4. Ablation: Validating Training–Inference Consistency

Table 3 studies the effect of training–inference alignment and the TBPTT truncation depth $K$ under our segment-level execution semantics. Here, *Misaligned* trains the model with standard full-context attention, but evaluates it under our segmented execution, resulting in mismatched forward semantics. Removing alignment (*Misaligned*) leads to a substantial performance drop across all LongBench(-E) categories, indicating the effect of optimizing a training objective consistent with inference execution. Notably, increasing the TBPTT depth from $K=1$ to $K=2$ does not improve performance and can slightly degrade it. This contrasts with classical recurrent models, and reflects a key difference in our framework, where the carried KV interface constitutes the only differentiable cross-segment state. These results empirically support $K=1$ as the most aligned and effective choice under our training–inference consistent segmented execution. Additional ablation results and analyses are provided in Appendix G.

## 5. Conclusion

We propose a training–inference consistent segmented execution framework for long-context language modeling under constrained execution. The framework treats segmented execution as a shared modeling assumption between training and inference, and aligns state evolution and credit assignment through a controlled cross-segment interface. Under this formulation, we show that when cross-segment recursion is strictly bounded, truncated backpropagation computes the exact gradient of a consistent training objective, preventing reliance on information unavailable at inference time. Empirically, the proposed approach performs competitively across diverse long-context benchmarks and achieves favorable latency–memory trade-offs, while substantially

improving scalability in extreme long-context regimes.

## Acknowledgements

This work was funded by National Natural Science Foundation of China (Grant No. 62366036), Outstanding Youth Fund Project of Inner Mongolia Autonomous Region (Grant No. 2025JQ010), Program for Young Talents of Science and Technology in Universities of Inner Mongolia Autonomous Region (Grant No. NJYT24033), Major Science and Technology Projects of Inner Mongolia Autonomous Region (Grant No. 2025ZDSF0029), Key R&D and Achievement Transformation Program of Inner Mongolia Autonomous Region (Grant No. 2025YFDZ0011, 2025YFDZ0026, 2025YFSH0021, 2025YFHH0073), Hohhot Science and Technology Project (Grant No. 2023-Zhan-Zhong-1).

## Impact Statement

This work improves the scalability of long-context language modeling by proposing a training–inference consistent segmented execution framework that reduces memory usage and inference latency while maintaining strong performance. The primary positive impact is enabling more accessible and energy-efficient deployment of long-context LLMs for applications such as document understanding and long-form summarization. As a general-purpose efficiency improvement, the method may also lower the cost of deploying long-context LLMs, which could amplify existing risks associated with misuse of such models, including misinformation generation or handling of sensitive content. We encourage responsible deployment through standard safeguards, including privacy-preserving data practices, access control for sensitive use cases, and content safety measures.

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

# A. Additional Theory: Exactness of TBPTT Under Consistent Segmented Objectives

## A.1. Computational-Graph Equivalence Viewpoint

This subsection elaborates the proof sketch in Theorem 3.3. Definition 3.2 inserts the stop-gradient boundary $\mathrm{sg}(C_{b_i})$ into the state chain, where $b_i = \max(0, i - K - 1)$. This eliminates any gradient path that traverses more than $K$ consecutive state transitions. Thus, for each loss term $\ell_i(\theta; \tilde{C}_{i-1}^{(K)}, R_{i-1})$, the backpropagation graph contains only the unrolled state updates from segments $b_i + 1$ to $i - 1$. TBPTT with truncation depth $K$ computes gradients on exactly this truncated graph, hence equals $\nabla_\theta L_K(\theta)$.

## A.2. Chain-Rule Derivation via Jacobian Products and Adjoint Recursion

We provide an explicit chain-rule expansion for the gradient computed by TBPTT and show that it equals $\nabla_\theta L_K(\theta)$.

**Notation.** Let $C_i \in \mathbb{R}^{d_C}$ be the vectorized interface state. Define the Jacobian of the state transition:

$$J_i := \frac{\partial C_i}{\partial C_{i-1}} \in \mathbb{R}^{d_C \times d_C}, \tag{6}$$

and the sensitivity of the transition to parameters:

$$U_i := \frac{\partial C_i}{\partial \theta}. \tag{7}$$

For a fixed loss term $\ell_i$, define its gradient w.r.t. the pre-segment state:

$$g_{i-1}^{(i)} := \frac{\partial \ell_i}{\partial C_{i-1}}. \tag{8}$$

**Truncated dependency induced by $\mathrm{sg}(\cdot)$.** Under $L_K(\theta)$, the computational graph for $\ell_i$ includes only the unrolled state transitions from $b_i + 1$ to $i - 1$, because $\tilde{C}_{b_i}^{(K)} = \mathrm{sg}(C_{b_i})$ blocks gradients into earlier segments. Therefore, for $\ell_i$ we only need to account for parameter effects on $C_j$ for $j \in \{b_i + 1, \ldots, i - 1\}$.

**Adjoint recursion (backward pass).** Define adjoints for $\ell_i$:

$$a_{i-1}^{(i)} := \frac{\partial \ell_i}{\partial C_{i-1}} = g_{i-1}^{(i)}, \tag{9}$$

$$a_{j-1}^{(i)} := a_j^{(i)} J_j, \quad \text{for } j = i - 1, i - 2, \ldots, b_i + 1. \tag{10}$$

This recursion is precisely the backward propagation across the truncated state chain, stopping at the boundary state $C_{b_i}$.

**Gradient expansion for one loss term.** By the chain rule, the gradient of $\ell_i$ w.r.t. $\theta$ under the truncated graph is

$$\frac{\partial \ell_i}{\partial \theta} = \left. \frac{\partial \ell_i}{\partial \theta} \right|_{\text{direct}} + \sum_{j=b_i+1}^{i-1} a_j^{(i)} U_j \tag{11}$$

where the "direct" term captures explicit dependence of $\ell_i$ on $\theta$ not mediated by $C$, and the summation captures all contributions flowing through the unrolled (and truncated) state updates.

**From one term to the full objective.** Summing Eq. (11) over $i$ yields

$$\nabla_\theta L_K(\theta) = \sum_{i=1}^{N} \left( \left. \frac{\partial \ell_i}{\partial \theta} \right|_{\text{direct}} + \sum_{j=b_i+1}^{i-1} a_j^{(i)} U_j \right), \tag{12}$$

which exactly matches the computation performed by TBPTT with truncation depth $K$: backpropagate each loss through at most $K$ transitions (Eq. (10)) and accumulate contributions.

This completes a constructive proof that TBPTT computes the exact gradient $\nabla_\theta L_K(\theta)$.

## B. Forward-Only Long-Range Retrieval Does Not Add Gradient Paths

**Lemma B.1** (No-gradient retrieval prefix). *Suppose the long-range prefix is constructed as*

$$R_{i-1} = \text{sg}\big(\rho(M_{i-1}, q_{i-1})\big), \tag{13}$$

*where $\rho$ is an arbitrary retrieval operator (e.g., top-k search), $M_{i-1}$ is an external memory, and $q_{i-1}$ is a query. Then $R_{i-1}$ introduces no gradient path w.r.t. model parameters:*

$$\frac{\partial R_{i-1}}{\partial \theta} = 0. \tag{14}$$

*Proof.* The stop-gradient operator $\text{sg}(\cdot)$ makes $R_{i-1}$ constant in backward propagation, hence its derivative w.r.t. $\theta$ is zero. □

**Implication.** Lemma B.1 implies the optional long-range module only changes forward conditioning but does not create additional cross-segment credit assignment paths. Therefore it does not affect Theorem 3.3.

## C. Attention Contexts for Local and Long-Range Heads

We formalize the per-head attention context used by the architecture in Section 3.4. Let segment $x^{(i)}$ have length $m_i$. For layer $\ell$ and head $h$, denote within-segment query/key/value matrices as $Q_i^{(\ell,h)} \in \mathbb{R}^{m_i \times d}$, $K_i^{(\ell,h)} \in \mathbb{R}^{m_i \times d}$, and $V_i^{(\ell,h)} \in \mathbb{R}^{m_i \times d}$.

We use standard scaled dot-product attention:

$$\text{Attn}(Q, K, V) = \text{softmax}\left(\frac{QK^\top}{\sqrt{d}} + \mathcal{M}\right) V, \tag{15}$$

where $\mathcal{M}$ is a causal mask over the concatenated prefix+segment sequence: each segment query position may attend to all prefix positions and to within-segment positions up to itself.

**Carried KV tail and retrieval prefix.** For local heads $h \in \mathcal{H}_{\text{local}}$, let the carried KV tail from the previous segment be $C_{i-1}^{(\ell,h)} = (K_{\text{carry}}^{(\ell,h)} \in \mathbb{R}^{M \times d}, V_{\text{carry}}^{(\ell,h)} \in \mathbb{R}^{M \times d})$. For long-range heads $h \in \mathcal{H}_{\text{long}}$ and $\ell \in \mathcal{L}_{\text{long}}$, let the retrieval prefix be $R_{i-1}^{(\ell,h)} = (K_{\text{ret}}^{(\ell,h)} \in \mathbb{R}^{R \times d}, V_{\text{ret}}^{(\ell,h)} \in \mathbb{R}^{R \times d})$.

**Context construction.** Define context keys/values $(K_{\text{ctx}}, V_{\text{ctx}})$ as:

$$(K_{\text{ctx}}, V_{\text{ctx}}) = \begin{cases} ([K_{\text{carry}}^{(\ell,h)}; K_i^{(\ell,h)}], [V_{\text{carry}}^{(\ell,h)}; V_i^{(\ell,h)}]) & \text{if } h \in \mathcal{H}_{\text{local}}, \\ ([K_{\text{ret}}^{(\ell,h)}; K_i^{(\ell,h)}], [V_{\text{ret}}^{(\ell,h)}; V_i^{(\ell,h)}]) & \text{if } h \in \mathcal{H}_{\text{long}} \text{ and } \ell \in \mathcal{L}_{\text{long}}, \\ (K_i^{(\ell,h)}, V_i^{(\ell,h)}) & \text{if } h \in \mathcal{H}_{\text{long}} \text{ and } \ell \notin \mathcal{L}_{\text{long}}. \end{cases} \tag{16}$$

The head output is then $O_i^{(\ell,h)} = \text{Attn}(Q_i^{(\ell,h)}, K_{\text{ctx}}, V_{\text{ctx}})$.

## D. Top-$k$ Retrieval Operator

We detail how the past-only KV pool produces a fixed-size retrieval prefix $R_{i-1}$ of length $R$.

**Memory layout.** For each $\ell \in \mathcal{L}_{\text{long}}$, we maintain a past-only KV pool storing key/value vectors for heads $h \in \mathcal{H}_{\text{long}}$ accumulated from earlier segments. Pool updates are append-only and performed without gradient flow (Lemma B.1).

**Query construction.** From segment $i-1$, we take the last $L_q$ query vectors from long-range heads and form a compact set of query summaries using (i) sliding-window mean pooling and (ii) a short tail average to emphasize recent tokens.

**Scoring and top-$k$.** For each head independently, we compute dot-product scores between query summaries and all keys in the pool, select top-$k$ indices, and deduplicate them. We then choose a small set of anchor indices with the highest scores.

**Window expansion and padding to length** $R$**.** We expand anchors with a small fixed offset window to obtain a contiguous set of indices, deduplicate/sort, and if fewer than $R$ indices remain we pad with a deterministic fallback (e.g., earliest positions) to reach exactly $R$ keys/values.

**Output.** We gather the selected keys/values as $(K_{\text{ret}}^{(\ell,h)}, V_{\text{ret}}^{(\ell,h)}) \in \mathbb{R}^{R \times d}$ and return them as the retrieval prefix $R_{i-1}^{(\ell,h)}$ used in Eq. (16).

## E. RoPE Re-indexing for Prefix Concatenation

We specify the RoPE positions used when concatenating a prefix of length $P$ before the current segment, where $P = M$ for the carried KV tail and $P = R$ for the retrieved prefix. For segment $i$ of length $m_i$, we assign prefix positions $p_{\text{pref}} = \{0, \ldots, P{-}1\}$ and segment positions $p_{\text{seg}} = \{P, \ldots, P{+}m_i{-}1\}$.

**RoPE application.** We apply RoPE to segment queries/keys using positions $p_{\text{seg}}$. For prefix KV (carried or retrieved), only keys require RoPE rotation; we apply RoPE to prefix keys using positions $p_{\text{pref}}$. Values are unchanged. This is equivalent to treating the prefix as preceding context under standard RoPE while leaving the segment attention structure unchanged.

## F. Additional Implementation Details

**Implementation details of our method.** Unless otherwise specified, our method is trained and evaluated under identical segment-level execution settings. Both training and inference use a fixed segment length of $S{=}4096$, and training samples are partitioned into non-overlapping segments at the individual sample level. The differentiable carried interface is instantiated as a KV tail of fixed length $M{=}512$ passed from segment $i{-}1$ to segment $i$. The long-range module is forward-only and uses the last $L_q \in [24, 64]$ query vectors from the previous segment to form retrieval queries, returning a fixed-size retrieved prefix of length $R{=}512$ when enabled.

For the main experiments, we use truncated backpropagation through time with truncation depth $K{=}1$, which corresponds to a maximum training context length of 8K tokens. Experiments with $K{=}2$ (maximum training context length 12K) are only used for ablation studies. For alignment fine-tuning, we optimize a standard language modeling objective on the SlimPajama dataset for 1,000 training steps, using a micro-batch size of 1 with gradient accumulation of 8. All alignment-based models are trained under identical optimization settings. Our primary experiments are conducted on LLaMA2-7B models with extended context lengths of 32K and 80K. Additional experiments on LLaMA-3.1 are reported only for LongBench-v2 evaluation.

**Implementation details of baseline methods.** We compare our approach against representative long-context baselines, including CCA-attention, MInference, NSA, StreamingLLM, and DuoAttention.

For CCA-attention, we adopt the official implementation and follow the default configuration, using a window size of 1024 and a group size of 16.

For MInference, we use the official implementation with the recommended sparse attention configuration provided by the authors.

For NSA (Yuan et al., 2025), since the official implementation was not publicly available at the time of our experiments, we use the reproduced implementation from Zhao et al. (2025). NSA is evaluated on the same LLaMA2-7B-80K backbone and under the same standard LongBench protocol used in Appendix G.1.

For StreamingLLM, the official implementation does not employ FlashAttention, which would lead to unfair efficiency comparisons in long-context settings. Therefore, we evaluate StreamingLLM through the implementation provided in the MInference framework, which supports FlashAttention-based execution. Specifically, we enable the StreamingLLM execution path by setting `attn_type=streaming` with `kv_type=streamingllm`, corresponding to a sliding-window attention mechanism with attention sinks. We use a local window size of 2044 tokens and retain 4 initial sink tokens (i.e., `n_local=2044` and `n_init=4`), following the standard configuration used in prior work.

For DuoAttention, we use the official implementation and the attention patterns released by the authors. For LLaMA2-7B-32K and LLaMA2-7B-80K, we apply the provided `Llama-2-7B-32K-Instruct` patterns. For LLaMA-3.1-8B-Instruct, we use the official `Meta-Llama-3.1-8B-Instruct` patterns. Unless otherwise specified, all baseline methods are

*Table 4.* Standard LongBench results on LLaMA2-7B-80K. LongBench-E is a length-stratified subset of LongBench; we additionally report results on the standard LongBench benchmark and include NSA as an additional sparse-attention baseline.

| Method | NQA | Qasper | MF-en | MF-zh | HQA | 2Wiki | MuSiQue | Dureader | GovReport | QMSum | MultiNews | TREC | TriviaQA | SAMSum | LCC | RepoBench-P | Avg. |
|---|---|---|---|---|---|---|---|---|---|---|---|---|---|---|---|---|---|
| LLaMA2-7B-80K | 2.07 | 4.99 | 6.64 | 3.59 | 6.00 | 4.15 | 2.32 | 17.97 | 10.55 | 20.34 | 8.56 | 69.00 | 86.75 | 43.29 | 55.66 | 45.33 | 24.20 |
| StreamingLLM | 1.03 | 3.06 | 4.93 | 1.88 | 6.83 | 2.31 | 2.81 | 8.10 | 5.03 | 15.30 | **12.04** | 60.50 | 87.07 | 41.36 | 57.32 | 45.86 | 22.21 |
| DuoAttention | 1.56 | 5.28 | 5.81 | 4.61 | 5.79 | 4.25 | 2.74 | 16.95 | **11.26** | 19.37 | 9.73 | 69.00 | 87.12 | 31.36 | 56.07 | 45.58 | 23.53 |
| MInference | 1.92 | 5.04 | 6.44 | 3.58 | 5.60 | 3.87 | 2.66 | **18.38** | 10.32 | **20.85** | 8.81 | **69.50** | 86.67 | **43.74** | 55.82 | 45.30 | 24.28 |
| NSA | 2.24 | 5.23 | 7.10 | 8.34 | 5.44 | 6.99 | 3.67 | 11.68 | 10.10 | 15.14 | 4.52 | 60.00 | 73.27 | 26.84 | 53.94 | 44.11 | 21.16 |
| Ours | **5.89** | **8.22** | **8.11** | **9.68** | **6.85** | **9.87** | **3.81** | 12.77 | 9.38 | 16.81 | 10.30 | 63.50 | **88.22** | 41.99 | **58.47** | **51.93** | **25.36** |

evaluated under their standard pretrained configurations without additional fine-tuning.

# G. Additional Experiment Results

## G.1. Standard LongBench Evaluation with NSA Baseline

As shown in Table 4, our method achieves the best overall average on the standard LongBench benchmark. Compared with NSA, our method obtains stronger average performance and improves substantially on QA-oriented tasks such as NarrativeQA, Qasper, HotpotQA, 2WikiMQA, and MuSiQue, suggesting that the gains observed on LongBench-E are not specific to the length-stratified subset.

## G.2. LLaMA-3.1 Results on LongBench-v2

Since the main experiments are conducted on LLaMA2-based backbones, we further examine whether the proposed method exhibits similar trends on a more recent architecture. We apply our approach to LLaMA-3.1-8B-Instruct and report additional results on LongBench-v2, without modifying the method or tuning hyperparameters for this backbone.

As shown in Table 5, our method shows performance trends consistent with those observed on LLaMA2, achieving competitive accuracy across difficulty levels and context length categories. While DuoAttention attains the highest overall score in this setting, our approach remains comparable without relying on backbone-specific attention patterns or additional fine-tuning, suggesting that the proposed training–inference consistent segmented execution generalizes beyond a specific backbone architecture.

*Table 5.* Comparisons on LongBench-V2.

| Method | Easy | Hard | Short | Medium | Long | Overall |
|---|---|---|---|---|---|---|
| Llama-3.1-8B-Instruct | 30.7 | 29.6 | 35.0 | 26.5 | 28.7 | 30.0 |
| StreamingLLM | 22.9 | 23.8 | 30.0 | 18.6 | 22.2 | 23.5 |
| MInference | 27.1 | 28.0 | 33.3 | 24.2 | 25.0 | 27.6 |
| DuoAttention | 31.2 | 31.2 | 36.1 | 27.0 | 31.5 | **31.2** |
| (Ours) | 27.6 | 31.2 | 33.3 | 28.4 | 26.9 | 29.8 |

## G.3. Effect of Training–Inference Alignment on Language Modeling Perplexity

To clarify why training–inference misalignment leads to the downstream performance degradation observed in the main experiments, we report language modeling perplexity under different TBPTT settings. Under aligned training–inference execution (TBPTT = 1 or 2), perplexity remains stable across evaluation context lengths, indicating that truncated back-propagation preserves language modeling stability in the proposed framework. In contrast, misaligned training results in rapidly increasing and unstable perplexity as context length grows, particularly beyond the training range. These results indicate that the downstream performance collapse reported in the main text is preceded by severe instability at the language modeling level, highlighting the necessity of training–inference consistency under constrained execution. This observation empirically corroborates the theoretical result that, under the proposed constrained cross-segment recursion, TBPTT with $K=1$ yields an exact gradient for the inference-consistent objective.

*Table 6.* Perplexity under increasing context lengths for different training–inference alignment and TBPTT settings.

| Method | 4K | 8K | 16K | 32K | 64K | Avg |
|---|---|---|---|---|---|---|
| Aligned (TBPTT=1) | 7.18 | 7.10 | 7.07 | 7.06 | 7.07 | 7.10 |
| Misaligned | 7.16 | 33.48 | 68.35 | 94.60 | 111.38 | 62.99 |
| Aligned (TBPTT=2) | 7.18 | 7.09 | 7.05 | 7.04 | 7.04 | **7.08** |

## G.4. Effect of Local State Capacity

To study the effect of local cross-segment state capacity, we evaluate language modeling perplexity and downstream long-context performance under different local KV state sizes, while keeping all other settings fixed. As shown in Table 7, increasing the local KV state size consistently reduces perplexity across context lengths, indicating improved stability of cross-segment modeling, with diminishing gains at larger capacities.

This trend is mirrored in downstream results on LongBench-E (Table 8). Compared to removing the local state, introducing a finite-capacity local KV state improves overall performance, while further increasing the state size yields only marginal and non-uniform gains across task categories. Overall, these results suggest that a moderate local state capacity is sufficient to balance language modeling stability and downstream performance under constrained execution. We hypothesize that overly large local state capacity may encourage over-reliance on short-range carried states, which does not uniformly benefit tasks requiring global abstraction, leading to non-monotonic gains across categories.

*Table 7.* Perplexity under increasing context lengths with different local state capacities.

| Local KV state size | 4K | 8K | 16K | 32K | 64K | Avg |
|---|---|---|---|---|---|---|
| 0 | 7.18 | 7.15 | 7.15 | 7.15 | 7.17 | 7.16 |
| 512 | 7.18 | 7.10 | 7.07 | 7.06 | 7.07 | 7.10 |
| 1024 | 7.18 | 7.07 | 7.03 | 7.03 | 7.03 | **7.07** |

*Table 8.* Effect of local state capacity on LongBench-E category performance.

| Local KV state size | S. QA | M. QA | Sum. | FS. Learning | Synthetic | Code | Avg |
|---|---|---|---|---|---|---|---|
| 0 | 7.10 | 7.35 | 7.58 | 62.37 | 0.35 | 54.88 | 23.27 |
| 512 | 7.58 | 8.56 | 8.70 | 63.76 | 0.42 | 56.02 | 24.17 |
| 1024 | 8.08 | 7.67 | 8.57 | 63.80 | 0.21 | 56.82 | **24.19** |

## G.5. Effect of Long-Range Module Placement

To examine the effect of long-range module placement, we compare language modeling perplexity and downstream long-context performance under different numbers of long-range layers. This ablation isolates the impact of long-range module depth while keeping all other settings unchanged.

From a language modeling perspective, perplexity remains highly similar across all configurations and evaluation context lengths (Table 9), indicating that inserting long-range modules does not materially affect basic language modeling stability. In contrast, downstream results on LongBench-E exhibit clearer differences (Table 10). Configurations without long-range modules or with only a small number of such layers achieve lower overall performance, whereas introducing long-range modules at more layers leads to consistent improvements, particularly on tasks requiring cross-segment information integration.

Taken together, these results indicate that long-range modules primarily improve downstream cross-segment reasoning without materially affecting language modeling perplexity, consistent with their architectural role as forward-only evidence access mechanisms that do not participate in state recursion.

*Table 9.* Perplexity under increasing context lengths with different numbers of long-range layers.

| Long-range layers | 4K | 8K | 16K | 32K | 64K | Avg |
|---|---|---|---|---|---|---|
| 0 | 7.17 | 7.10 | 7.06 | 7.04 | 7.04 | **7.08** |
| 2 | 7.18 | 7.10 | 7.06 | 7.05 | 7.05 | 7.09 |
| 4 | 7.18 | 7.10 | 7.07 | 7.06 | 7.07 | 7.10 |

*Table 10.* LongBench-E category performance with different numbers of long-range layers.

| Long-range layers | S. QA | M. QA | Sum. | FS. Learning | Synthetic | Code | Avg |
|---|---|---|---|---|---|---|---|
| 0 | 6.10 | 5.52 | 8.77 | 62.10 | 0.00 | 53.29 | 22.63 |
| 2 | 5.47 | 5.61 | 7.51 | 62.51 | 0.08 | 53.47 | 22.44 |
| 4 | 7.58 | 8.56 | 8.70 | 63.76 | 0.42 | 56.02 | **24.17** |

### G.6. Effect of Long-Range Head Grouping

We further ablate the choice of long-range heads while keeping the number of long-range heads fixed. All variants use the same long-range-enabled layers $\mathcal{L}_{\mathrm{long}} = \{6, 8, 11, 18\}$ and differ only in how attention heads are assigned to $\mathcal{H}_{\mathrm{long}}$ and $\mathcal{H}_{\mathrm{local}}$. We compare three head grouping strategies: (i) contiguous grouping, where $\mathcal{H}_{\mathrm{long}} = \{0, \ldots, 15\}$ and $\mathcal{H}_{\mathrm{local}} = \{16, \ldots, 31\}$; (ii) interleaved grouping, where $\mathcal{H}_{\mathrm{long}} = \{0, 2, 4, \ldots, 30\}$ and $\mathcal{H}_{\mathrm{local}} = \{1, 3, 5, \ldots, 31\}$; and (iii) prior-based grouping, which uses $\mathcal{H}_{\mathrm{long}} = \{0, 1, 2, 4, 9, 12, 14, 15, 16, 18, 19, 22, 23, 26, 29, 30\}$. The remaining heads are assigned to $\mathcal{H}_{\mathrm{local}}$. All layer and head indices follow the zero-indexed implementation convention.

As shown in Table 11, contiguous and interleaved groupings obtain comparable average performance, while the prior-based grouping achieves the best overall average. This suggests that the benefit of the long-range channel depends not only on the number of long-range heads, but also on assigning long-range capacity to heads with stronger retrieval-oriented behavior.

*Table 11.* Effect of long-range head grouping on the standard LongBench benchmark using LLaMA2-7B-80K. All variants use the same long-range-enabled layers $\mathcal{L}_{\text{long}} = \{6, 8, 11, 18\}$ and the same number of long-range heads; only the head grouping strategy is changed.

| Head Grouping | NQA | Qasper | MF-en | MF-zh | HQA | 2Wiki | MuSiQue | Dureader | GovReport | QMSum | MultiNews | TREC | TriviaQA | SAMSum | LCC | RepoBench-P | Avg. |
|---|---|---|---|---|---|---|---|---|---|---|---|---|---|---|---|---|---|
| Contiguous | 4.18 | 4.75 | 6.88 | 5.99 | 6.27 | 5.07 | 3.11 | 9.62 | **11.51** | 16.18 | **10.92** | 62.00 | 86.32 | **42.18** | 56.27 | 51.89 | 23.95 |
| Interleaved | 5.32 | 5.04 | 6.74 | 6.09 | 5.61 | 7.15 | 2.77 | 10.31 | 6.59 | 14.89 | 10.37 | 62.50 | 87.68 | 41.67 | 58.14 | **52.17** | 23.94 |
| Prior-based | **5.89** | **8.22** | **8.11** | **9.68** | **6.85** | **9.87** | **3.81** | **12.77** | 9.38 | **16.81** | 10.30 | **63.50** | **88.22** | 41.99 | **58.47** | 51.93 | **25.36** |

