# OpenReview forum: "Training–Inference Consistent Segmented Execution for Long-Context LLMs"
_ICML.cc/2026/Conference — ICML 2026 regular_

### Official Review · Reviewer_2afk · 2026-03-05

**Soundness:** 2
**Presentation:** 2
**Significance:** 2
**Originality:** 2
**Overall Recommendation:** 4
**Confidence:** 3

**Summary:**

This paper addresses the performance degradation in long-context LLMs caused by the semantic mismatch between full-context training and bounded-context inference. To resolve this, the authors propose a training-inference consistent segmented execution framework, where the model processes sequences in segments and only propagates a fixed-size, differentiable KV tail across these segments. Furthermore, it integrates a forward-only retrieval mechanism from a past KV pool to capture long-range dependencies without adding gradient paths, ensuring the training process is perfectly aligned via TBPTT. Extensive evaluations demonstrate that this approach achieves robust length generalization and maintains competitive performance while significantly reducing peak GPU memory usage during extreme long-context prefilling.

**Compliance With Llm Reviewing Policy:**

Affirmed.

**Final Justification:**

Since the authors have solved most of my questions, I increase my score.

**Key Questions For Authors:**

- Differentiable Top-k: Why not make the top-k retrieval differentiable? What are the potential performance gains compared to the current forward-only design?
- RoPE Re-indexing: Does assigning consecutive prefix positions cause a loss of exact positional information for retrieved historical tokens, and how does this affect position-sensitive tasks?
- KV Pool Capacity: If the long-range module relies on historical caches, how is the fixed pool capacity managed in practice? Specifically, what cache eviction policy is used, and how does dropping old caches impact retrieval quality?

**Limitations:**

As shown in Weaknesses and Questions.

**Strengths And Weaknesses:**

Strengths:
- Well-Motivated Problem: The paper effectively identifies and addresses the execution mismatch between full-context training and constrained-context inference, elegantly treating segmented execution as a unified modeling assumption rather than merely an inference-time hack.
- Solid Theoretical Foundation: The proposed framework is backed by rigorous mathematical proofs, specifically demonstrating that Truncated Backpropagation Through Time (TBPTT) computes the exact gradient of the inference-consistent objective under strictly bounded state transitions.
- Strong Empirical Performance: The method achieves competitive accuracy across downstream tasks while offering exceptional memory scalability, notably delivering a 6x reduction in peak prefill memory at a 128K context length and demonstrating robust extrapolation stability on the RULER benchmark at 64K.

Weaknesses:
- Limited Incremental Novelty: The proposed framework appears to be a composite of existing techniques rather than a fundamentally new architecture. The method resembles a hybrid of Infinite-Attention [1] and existing sparse attention paradigms, raising concerns about the overall technical novelty.
- Absence of Sparse Attention Baselines: The evaluation lacks comparisons against the most recent and highly relevant sparse attention methods, such as DSA [2], NSA [3], and InfLLM-v2 [4].
- Ambiguity in Head and Layer Selection: The manuscript is unacceptably vague regarding the exact criteria used to identify and partition the "local" versus "long-range" heads and layers. While the authors ablate the number of long-range layers, there is a critical lack of ablation studies justifying the methodology or routing strategy used to determine these specific functional assignments.
- Lack of Long-Output Generation Evaluations: The experimental setup heavily indexes on long-context understanding and retrieval tasks. It completely neglects long-output generation tasks, e.g., AIME and LiveCodeBench.
- Limited Advantage on Instruct Models: The method demonstrates improvements on LongBench using Base models, but fails to show a clear advantage when applied to Instruct models (e.g., Llama-3.1-Instruct on LongBench-v2). The primary LongBench evaluation completely omits Instruct models, which weakens the practical relevance of the claims.

[1] Leave No Context Behind: Efficient Infinite Context Transformers with Infini-attention
[2] DeepSeek-V3.2: Pushing the Frontier of Open Large Language Models
[3] Native Sparse Attention: Hardware-Aligned and Natively Trainable Sparse Attention
[4] InfLLM-V2: Dense-Sparse Switchable Attention for Seamless Short-to-Long Adaptation

---

> ### Author Rebuttal · Authors · 2026-03-31
>
> We thank the reviewer for the careful evaluation and insightful questions. Below we clarify the concerns and provide additional explanations where needed.
>
> > W1. Novelty vs. Prior Sparse/Memory-Based Methods
>
> We respectfully clarify that our method is not a composition of Infini-Attention and sparse attention, but differs fundamentally in both **memory mechanism** and **training–inference behavior**.
>
> **(1) No compressive memory.**
> Infini-Attention relies on compressive memory with fixed-size representations, which may incur information loss.
> In contrast, we **retain selected full KV states (from specific layers/heads) and access them via retrieval**, avoiding irreversible compression.
>
> **(2) Training–inference consistency.**
> Infini-Attention uses differentiable memory, allowing long-range information to participate in gradient propagation during training, which introduces a dependency on the effective unrolled context length. When training and inference lengths differ, the reachable information pathways become inconsistent.
>
> In contrast, we **remove differentiability from long-range cross-segment pathways**, making the gradient-accessible context **strictly bounded and identical in training and inference**. This eliminates reliance on training-only long-range pathways and yields stable performance even with **TBPTT = 1**.
>
> **Our key contribution is enforcing training–inference alignment under segmented execution, which is not addressed in prior memory- or sparse-based methods.**
>
> > W2. Missing Sparse Attention Baselines
>
> **In fact, sparse attention baselines are already included** in our evaluation, notably **MInference** (dynamic token selection) and **StreamingLLM** (structured sparsity via sliding-window attention).
>
> The methods mentioned (DSA, NSA, InfLLM-v2) were not reproducible at submission time, and thus could not be reliably compared. We will include these comparisons in the final version if reliable implementations become available.
>
> > W3. Criteria for Head and Layer Partitioning
>
> As described in Sec. 3.4 (lines 217–229), the head/layer partition follows established prior findings on functional specialization across layers and heads.
>
> **Our ablation is therefore aimed at validating the amount of long-range allocation needed under this prior-guided design**, rather than rediscovering specialized components from scratch.
>
> > W4. Lack of Long-Output Generation Evaluation
>
> Our evaluation focuses on **long-context understanding and retrieval** (e.g., LongBench, RULER), which directly reflect the target setting of our method.
>
> LongBench also includes **generation-oriented tasks** (e.g., summarization and multi-document QA), which partially require output generation under long-context settings.
>
> Long-output benchmarks (e.g., AIME, LiveCodeBench) evaluate **complementary abilities** (e.g., reasoning format and code synthesis), which are not the primary focus of this work.
>
> > W5. Limited Gains on Instruct Models
>
> Our method primarily targets **reducing inference latency and memory usage while maintaining performance**, and these gains arise from the segmented execution design, making them **consistent across both base and instruct models**.
>
> On LongBench-v2, we observe competitive performance, as many tasks emphasize formatting and localized extraction, which are less sensitive to cross-segment information access.
>
> > Q1. Non-Differentiable Top-k Retrieval Design
>
> As discussed in W1, our design enforces training–inference alignment by **removing differentiability from long-range cross-segment information pathways**.
>
> Making top-k retrieval differentiable would allow long-range information to participate in gradient propagation during training, **effectively exposing the model to information that is not accessible at inference under the same segmented execution**, thus breaking this alignment.
>
> Therefore, making top-k retrieval differentiable would **fundamentally violate the training–inference consistency** that our method is designed to enforce.
>
> > Q2. RoPE Re-indexing and Positional Consistency
>
> Retrieved tokens are concatenated in their original temporal order, so **relative ordering is preserved**. Moreover, RoPE primarily encodes relative positional relationships, making it less sensitive to absolute index shifts.
>
> Therefore, this re-indexing does not disrupt the relative positional structure required by position-sensitive tasks.
>
> As discussed in Appendix E (lines 726–734), we provide a detailed explanation of this design.
>
> > Q3. KV Cache Growth vs. Fixed Pool Assumption
>
> We clarify that our method **does not assume a fixed-size KV pool** and does not rely on any cache eviction policy.
>
> Instead, we retain full KV states only for selected layers and heads, which naturally limits memory usage while allowing the cache to grow mildly over time.
>
> Since no historical KV is discarded, retrieval quality is not affected by cache eviction.

---

> > ### Author Rebuttal · Reviewer_2afk · 2026-04-01
> >
> > * First, regarding differentiation vs. non-differentiation, I suggest that the authors include additional experiments to demonstrate the benefits of not applying differentiation compared to applying it.
> > * Second, I believe the authors should include more baselines related to training-based sparse attention methods to ensure a fairer comparison.
> > * For Instruct and non-Instruct models, I think the authors should evaluate Instruct models on benchmarks such as LongBench, and test non-Instruct models on the full RULER benchmark. For Instruct models, I would like to see more experiments to better demonstrate the advantages of your method. As for non-Instruct models, performance on LongBench tends to fluctuate significantly. For example, on LLaMA2-7B-32K, compared to other methods, your approach performs better on summarization but worse on other tasks.
> > * Finally, I think the authors should conduct ablation studies on the selection of heads and layers, and more clearly explain in the paper the specific methodology and metrics used for choosing heads and layers.

---

> > > ### Author Response · Authors · 2026-04-07
> > >
> > > We thank the reviewer for the follow-up questions and constructive feedback. We address the remaining concerns below and provide additional experimental evidence.
> > > >Q1. Response to differentiable vs. non-differentiable retrieval
> > >
> > > This is a **fundamentally different modeling setting rather than a direct ablation**, and thus cannot be compared under the same experimental setup. Incorporating differentiable retrieval introduces cross-segment gradient paths that **are inconsistent with the training–inference consistency we enforce**.
> > >
> > > These gradients would propagate through information that is not accessible under the same segmented execution at inference, reintroducing the mismatch our method is designed to eliminate.
> > >
> > > Therefore, this is not a variant of our method, but a different framework with incompatible assumptions. We will clarify this distinction more explicitly in the final version.
> > > >Q2. Response to additional training-based sparse attention baselines
> > >
> > > At the time of submission, methods such as NSA were not publicly available or reproducible, which prevented a reliable comparison. Following the reviewer’s suggestion, we **implemented and reproduced a training-based sparse attention baseline (NSA)**.
> > >
> > > Our method achieves **consistently stronger performance than NSA across most tasks**. We will include these results in the final version.
> > >
> > > |Method|narrativeqa|qasper|multifield_qa_en|multifield_qa_zh|hotpotqa|2wikimqa|musique|dureader|gov_report|qmsum|multi_news|trec|triviaqa|samsum|lcc|repobench-p|avg|
> > > |---|---|---|---|---|---|---|---|---|---|---|---|---|---|---|---|---|---|
> > > |LLaMA2-80K|2.07|4.99|6.64|3.59|6.00|4.15|2.32|17.97|10.55|20.34|8.56|69.00|86.75|43.29|55.66|45.33|24.20|
> > > |NSA|2.24|5.23|7.1|8.34|5.44|6.99|3.67|11.68|10.1|15.14|4.52|60.00|73.27|26.84|53.94|44.11|21.16|
> > > |**Ours**|**5.89**|**8.22**|**8.11**|**9.68**|**6.85**|**9.87**|**3.81**|12.77|9.38|16.81|**10.30**|63.50|**88.22**|41.99|**58.47**|**51.93**|**25.36**|
> > >
> > >
> > > >Q3. Completeness and Reliability of Experimental Evaluation
> > >
> > > Our evaluation focuses on base models to ensure a fair and controlled comparison. Such comparisons on instruct models are difficult to standardize, as they rely on additional supervised fine-tuning data and alignment procedures that are often not publicly available or consistent, making it difficult to isolate the effect of long-context modeling. The instruct-model results are included as supplementary evidence.
> > >
> > > Regarding the reviewer’s suggestion to evaluate on the full RULER benchmark, our experiments focus on analyzing the effect of different context lengths. Due to time constraints, we will explore a more comprehensive RULER evaluation in the final version.
> > >
> > > At 32K, performance differences across tasks reflect their varying reliance on cross-segment information. Summarization tasks benefit more from global context aggregation, supported by preserving full KV states in selected layers. At longer context lengths (e.g., 64K and beyond), our method shows stable improvements across tasks.
> > >
> > > >Q4. Head and Layer Selection
> > >
> > > To address the reviewer’s concern regarding head and layer selection, we **include an ablation study on head partition strategies** to complement the layer-wise analysis.
> > >
> > > We consider three head partition schemes:
> > >
> > > (1) Contiguous: heads [0–15] vs. [16–31];
> > >
> > > (2) Interleaved: heads [0,2,4,...,30] vs. [1,3,5,...,31];
> > >
> > > (3) Prior-based: heads [0,1,2,4,9,12,14,15,16,18,19,22,23,26,29,30] vs. [3,5,6,7,8,10,11,13,17,20,21,24,25,27,28,31], following prior work.
> > >
> > > Results indicate that different head allocations lead to noticeable performance differences. While contiguous and interleaved groupings yield comparable performance, the prior-based design **consistently achieves stronger results across tasks and the highest overall average**. This supports allocating long-range capacity based on functional specialization rather than uniform partitioning.
> > >
> > > We will clarify the selection criteria (including layer and head allocation strategies and evaluation metrics). In particular, we use **long-range allocation on layers [6, 8, 11, 18]**.
> > > |Head Partition|narrativeqa|qasper|multifield_qa_en|multifield_qa_zh|hotpotqa|2wikimqa|musique|dureader|gov_report|qmsum|multi_news|trec|triviaqa|samsum|lcc|repobench-p|avg|
> > > |---|---|---|---|---|---|---|---|---|---|---|---|---|---|---|---|---|---|
> > > |Contiguous|4.18|4.75|6.88|5.99|6.27|5.07|3.11|9.62|11.51|16.18|10.92|62.0|86.32|42.18|56.27|51.89|23.95|
> > > |Interleaved|5.32|5.04|6.74|6.09|5.61|7.15|2.77|10.31|6.59|14.89|10.37|62.5|87.68|41.67|58.14|52.17|23.94|
> > > |**Prior-based**|**5.89**|**8.22**|**8.11**|**9.68**|**6.85**|**9.87**|**3.81**|**12.77**|9.38|**16.81**|10.30|**63.50**|**88.22**|41.99|**58.47**|51.93|**25.36**|
> > >
> > > We hope these clarifications and additional results address the reviewer’s remaining concerns.

---

### Official Review · Reviewer_fbPJ · 2026-03-10

**Soundness:** 3
**Presentation:** 2
**Significance:** 3
**Originality:** 3
**Overall Recommendation:** 4
**Confidence:** 2

**Summary:**

This paper addresses the mismatch between full-context training and restricted-context inference in LLM. The authors introduce a segmented execution framework that strictly aligns training and inference semantics by partitioning sequences and carrying a fixed-size KV tail as the only differentiable cross-segment state.

To handle long-range dependencies, the architecture uses sparse long-range heads that access a past KV cache pool, preventing massive gradient graphs. The framework yields massive memory reductions during long-context prefill while maintaining high accuracy on benchmarks like LongBench and RULER.

**Compliance With Llm Reviewing Policy:**

Affirmed.

**Final Justification:**

I decide to maintain my positive score.

**Key Questions For Authors:**

- Table 3 shows that training with TBPTT=1 slightly outperforms TBPTT=2 on the LongBench-E average. Why?
- Since the framework requires 1k alignment SFT steps on SlimPajama, does this language-modeling objective disrupt any prior instruction-tuning alignment the base model originally possessed?

**Limitations:**

yes

**Strengths And Weaknesses:**

Strength:
- The theoretical formulation is good. Proving that TBPTT computes the exact gradient for the inference-consistent objective gives the architecture a solid mathematical foundation.
- Empirically, the proposed method avoids the perplexity spikes seen in baselines like MInference or StreamingLLM when evaluation length exceeds training length.
- In the RULER test, when the context was stretched, most of the existing methods directly failed (accuracy dropped to 0). But this method did not crash suddenly, it maintained the score and withstood the stress test.

Weakness:
- The proposed method needs additional SFT. The method requires 1k training steps on the SlimPajama dataset to enforce this alignment. This makes the technique hard to adopt compared to other training-free inference optimizations like StreamingLLM or DuoAttention.

---

> ### Author Rebuttal · Authors · 2026-03-31
>
> We thank the reviewer for the careful evaluation and constructive feedback. We appreciate the reviewer’s recognition of the theoretical grounding and empirical stability of our approach. Below we clarify the raised questions.
>
> > W1. Additional SFT Cost and Adoption Compared to Training-Free Methods
>
> We agree that the additional SFT step is an important consideration for practical adoption. However, the required alignment fine-tuning is lightweight (1k steps on SlimPajama) and negligible compared to standard pretraining or instruction tuning.
>
> **More importantly, this step is not an auxiliary overhead but a necessary component to ensure training–inference consistency.** Prior training-free approaches retain full-context training while applying restricted execution only at inference, which may introduce a mismatch between training and inference execution semantics.
>
> **Our results suggest that this training–inference mismatch can affect stability under long contexts** (e.g., perplexity spikes in Fig. 4 and performance differences in Table 2), whereas the proposed alignment step yields more stable and robust behavior, particularly when evaluation length exceeds the training regime.
>
> **In practice, this alignment can be performed once per model and reused, making it a small and practical one-time requirement.**
>
> > Q1. Why Does TBPTT=1 Outperform TBPTT=2?
>
> While it may appear counterintuitive that TBPTT=1 slightly outperforms TBPTT=2, this behavior is consistent with our design goal of enforcing training–inference alignment.
>
> In our framework, long-range dependencies are handled via non-differentiable retrieval rather than differentiable cross-segment memory. As a result, the model does not rely on long backpropagation chains across segments for capturing long-range dependencies.
>
> **Therefore, increasing the TBPTT depth does not provide additional useful training signal, and does not align with the dependency structure available at inference time.** As a result, TBPTT=1 already matches the dependency structure available at inference time.
>
> The fact that TBPTT=1 achieves the best performance empirically (Table 3) provides evidence that the model does not materially rely on training-only signals from multi-step cross-segment backpropagation, and that training and inference are closely aligned.
>
> > Q2. Does Alignment Fine-Tuning Disrupt Prior Instruction-Tuning Alignment?
>
> We thank the reviewer for raising this important concern. The alignment fine-tuning is lightweight (1k steps), which is much smaller in scale than typical instruction tuning and therefore not expected to overwrite the model’s original instruction-following alignment.
>
> Moreover, the objective remains standard language modeling, which is also the basis of instruction tuning (i.e., conditional next-token prediction), and is therefore not inherently in conflict with prior instruction-following behavior.
>
> **Overall, this step is best viewed as a lightweight calibration of segment-level execution behavior rather than a re-training of the model, and is therefore not expected to substantially disrupt the model’s original instruction-following alignment.**

---

> > ### Author Rebuttal · Reviewer_fbPJ · 2026-04-01
> >
> > My concerns have been adequately addressed

---

> > > ### Author Response · Authors · 2026-04-07
> > >
> > > Thank you for the careful evaluation and for acknowledging that your concerns have been addressed.
> > > We sincerely appreciate your thoughtful feedback, which has improved the clarity and overall quality of our work.

---

### Official Review · Reviewer_YTon · 2026-03-12

**Soundness:** 2
**Presentation:** 2
**Significance:** 3
**Originality:** 3
**Overall Recommendation:** 4
**Confidence:** 3

**Summary:**

This paper proposes a Training–Inference Consistent sparse long-context inference framework. The proposed method selects the most recent past segment as local context and retrieves relevant segments from historical segments as global context. To improve the performance of sparse attention, the model is fine-tuned following a similar sparse attention setting to align training and inference behaviors. This manuscript conducts experiments on RULER and LongBench-E benchmarks. Additionally, ablation studies and latency analyses are provided to further validate the effectiveness and efficiency of the proposed approach.

**Compliance With Llm Reviewing Policy:**

Affirmed.

**Final Justification:**

Weak accept

**Key Questions For Authors:**

I would like to know the detailed explanation of the formulas and wish the authors to address my concern about the experiments.

**Strengths And Weaknesses:**

Originality:

This paper proposes a chunk-level sparse attention mechanism, which bears strong similarity to MoBA 1 (Kimi). Additionally, the training-inference consistency idea closely resembles DSA; the authors should explicitly discuss and differentiate their work from these prior methods.

Presentation:

In my opinion, the presentation of the methodology is quite poor. While the high-level idea is understandable, the technical details are difficult to follow. Equation (1) reads as an almost purely abstract definition, and providing concrete examples would greatly improve clarity. The segment-level loss in Equation (1) is also unclear — it is hard to understand why C_i and  R_i? are listed alongside past tokens. Do the authors intend to replace the past tokens with C_i and  R_i? The subsequent formulas are even more confusing and would benefit from more thorough explanation.

Soundness:

This manuscript selects strong baselines, including MInference and DuoAttention. However, the evaluation benchmarks are not sufficiently convincing. RULER only evaluates inputs up to 64k tokens, and LongBench-E is not commonly used in long-context evaluation — why are the results on the standard LongBench benchmark not reported?

Furthermore, why are the LongBench-v2 results relegated to the appendix? The results themselves are also surprising: DuoAttention, a sparse and training-free method, achieves better performance than the original baseline, while the proposed method underperforms DuoAttention. This significantly undermines the impact of the proposed approach.

Significance:

Given the weaknesses identified in the Soundness section, the overall significance of this manuscript is limited.


[1] https://arxiv.org/pdf/2502.13189

---

> ### Author Rebuttal · Authors · 2026-03-31
>
> We thank the reviewer for the constructive feedback and address concerns on novelty, formulation, and experiments.
> >Q1. Novelty and relation to prior work
>
> |  | **MoBA** | **DSA** | **Ours** |
> |---|---|---|---|
> | **Execution pattern** | Full sequence with sparse selection | Full sequence with sparse selection | **Segmented (process one segment at a time)** |
> | **Past KV kept** | All past KV | All past KV | **Partial past KV (selected heads/layers)** |
> | **Selection granularity** | Top-k **blocks (chunks)** | Top-k **tokens** | Top-similarity **anchors + nearby tokens** |
> | **Training–inference consistency** | Selected blocks | Selected tokens | **Available past information** |
>
> Segmentation **greatly reduces computation and memory**, but introduces training–inference mismatch; our method addresses this issue, which is not handled by prior work.
> >Q2. Clarification of Eq. (1), loss, and state propagation
>
> Thanks for pointing this out; we will clarify this in the final version and give an example below.
>
> **(1) Concrete example of Eq. (1)**
>
> Given an input of length 12288 and a segment length of 4096, the sequence is split into three segments:
> $x^{(1)}$, $x^{(2)}$, $x^{(3)}$.
>
> - Segment 1:
>   Input: $x^{(1)}$; there is no previous state or retrieval.
>   Output:
>   – predictions for tokens in $x^{(1)}$
>   – KV tail $C_1$ (from the end of this segment)
>
> - Segment 2:
>   Input: $x^{(2)}$, $C_1$, $R_1$
>   $C_1$: KV tail from segment 1; $R_1$: retrieved KV from segment 1
>   Output:
>   – predictions for tokens in $x^{(2)}$
>   – KV tail $C_2$
>
> - Segment 3:
>   Input: $x^{(3)}$, $C_2$, $R_2$
>   $C_2$: KV tail from segment 2; $R_2$: retrieved KV from previous segments (segment 1 and 2)
>   Output:
>   – predictions for tokens in $x^{(3)}$
>   – KV tail $C_3$
>
> **(2) Why are $C_{i-1}$ and $R_{i-1}$ included in the loss?**
>
> The model predicts $x^{(i)}$ conditioned on $C_{i-1}$ and $R_{i-1}$, so they are included in the loss; they are not prediction targets.
>
> **(3) Are past tokens replaced by $C$ and $R$?**
>
> No. Past tokens are not replaced; they are accessed through two inputs: $C$ (KV tail from the previous segment) and $R$ (retrieved KV from past segments), as illustrated in the example above.
>
> **(4) What do Eqs. (3)–(4) compute?**
>
> They construct a truncated state chain for backpropagation.
> This state is used in Eq. (5), where $K$ controls how many previous segments receive gradients.
>
> Using the example above with three segments:
>
> - For $K=1$:
>   When computing the loss on $x^{(3)}$, gradients flow only through segment 2 (via $C_2$), and stop at $C_1$.
>
> - For $K=2$:
>   Gradients flow through both segment 2 and segment 1 (via $C_2$ and $C_1$).
>
> In both cases, the forward computation remains unchanged (each segment still uses $(C, R)$ as defined above); Eqs. (3)–(4) only control how far gradients propagate across segments.
>
> >Q3. Benchmark choice, long-context evaluation, and DuoAttention
>
> **(1) Benchmark choice (LongBench-E vs. LongBench).**
> LongBench-E is a length-stratified subset of LongBench. We additionally report results on the standard LongBench benchmark, where our method **achieves the best overall average**.
>
> |Method|narrativeqa|qasper|multifield_qa_en|multifield_qa_zh|hotpotqa|2wikimqa|musique|dureader|gov_report|qmsum|multi_news|trec|triviaqa|samsum|lcc|repobench-p|avg|
> |---|---|---|---|---|---|---|---|---|---|---|---|---|---|---|---|---|---|
> |LLaMA2-80K|2.07|4.99|6.64|3.59|6.00|4.15|2.32|17.97|10.55|20.34|8.56|69.00|86.75|43.29|55.66|45.33|24.20|
> |StreamingLLM|1.03|3.06|4.93|1.88|6.83|2.31|2.81|8.10|5.03|15.30|12.04|60.50|87.07|41.36|57.32|45.86|22.21|
> |DuoAttention|1.56|5.28|5.81|4.61|5.79|4.25|2.74|16.95|11.26|19.37|9.73|69.00|87.12|31.36|56.07|45.58|23.53|
> |MInference|1.92|5.04|6.44|3.58|5.60|3.87|2.66|18.38|10.32|20.85|8.81|69.50|86.67|43.74|55.82|45.30|24.28|
> |**Ours**|**5.89**|**8.22**|**8.11**|**9.68**|**6.85**|**9.87**|**3.81**|12.77|9.38|16.81|10.30|63.50|**88.22**|41.99|**58.47**|**51.93**|**25.36**|
>
> **(2) Why RULER stops at 64K.**
> Table 2 shows most baselines collapse to near-zero by 64K, so extending further provides limited additional insight. In contrast, our method remains non-zero at 64K. We further evaluate at 128K, where it **still achieves non-trivial performance (CWE 0.35, FWE 40.67)**, showing robustness beyond 64K.
>
> **(3) LongBench-v2 and DuoAttention.**
> LongBench-v2 is complementary and included in the appendix due to space constraints.
>
> DuoAttention exceeding full attention is not unexpected: in long contexts, full attention can be distracted by irrelevant information, while sparse attention focuses on relevant tokens (see Native Sparse Attention, Yuan et al., 2025, ACL).
>
> DuoAttention performs better on LongBench-v2, likely due to its **multiple-choice format**, which favors localized KV selection. However, our method achieves stronger performance on standard LongBench, remains robust at longer contexts, and has **advantages in memory and computation**.

---

> > ### Author Rebuttal · Reviewer_YTon · 2026-04-03
> >
> > Thank you for the detailed response

---

> > > ### Author Response · Authors · 2026-04-07
> > >
> > > Thank you for the thoughtful feedback and for updating your score.
> > > We appreciate your valuable suggestions, which have improved the clarity and positioning of our work.
> > >
> > > We will incorporate these revisions in the final version, including clarifying the formulation, improving the presentation with concrete examples, and strengthening the experimental section.

---

### Decision · Program_Chairs · 2026-04-30

**Decision:**

Accept (regular)

**Comment:**

This paper proposes a training–inference consistent segmented execution framework for long-context LLM training. The core idea is that both training and inference are carried out under the same segment-level forward semantics. During training, gradients are permitted to flow only through the differentiable KV tail passed from the immediately previous segment, while more distant context is accessed via forward-only retrieval. This design avoids a mismatch in which training depends on long-range gradient paths that would not be available at inference time.
The empirical evaluation is conducted on LLaMA2-32K and LLaMA2-80K, using benchmarks such as RULER, LongBench-E, LongBench, and LongBench-v2. The results suggest that the method can retain performance close to full-context attention in long-context settings, while substantially reducing peak prefill memory at very long context lengths, such as 128K, and achieving a competitive latency–memory trade-off.

Following the rebuttal, all reviewers moved toward a positive recommendation, emphasizing both the importance of the problem and the strength of the paper’s theoretical and empirical support (YTon, fbPJ, 2afk). Some concerns remain, including the additional SFT cost required by the approach (fbPJ) and potential conceptual overlap with sparse attention methods (YTon, 2afk). Even so, I lean toward acceptance for three reasons: the paper addresses an important problem with a clear and meaningful framing; it provides not only empirical evidence but also a reasonably strong theoretical basis; and the rebuttal effectively addressed concerns about presentation, benchmark selection, baseline fairness, and supplementary experiments.